# The Changes Occurring in Proteins during Processing and Storage of Fermented Meat Products and Their Regulation by Lactic Acid Bacteria

**DOI:** 10.3390/foods11162427

**Published:** 2022-08-12

**Authors:** Daixun Wang, Feng Cheng, Yi Wang, Jun Han, Fang Gao, Jianjun Tian, Kaiping Zhang, Ye Jin

**Affiliations:** 1College of Food Science and Engineering, Inner Mongolia Agricultural University, Hohhot 010018, China; 2Ministry of Agriculture and Rural Affairs Integrative Research Base of Beef and Lamb Processing Technology, Hohhot 010018, China; 3Department of Cooking & Food Processing, Inner Mongolia Business and Trade Vocational College, Hohhot 010070, China

**Keywords:** fermented meat products, protein, degradation, oxidation, lactic acid bacteria, antioxidation

## Abstract

Protein, which is the main component of meat, is degraded and oxidized during meat fermentation. During fermentation, macromolecular proteins are degraded into small peptides and free amino acids, and oxidation leads to amino acid side chain modification, molecular crosslinking polymerization, and peptide chain cleavage. At different metabolic levels, these reactions may affect the protein structure and the color, tenderness, flavor, and edible value of fermented meat products. Lactic acid bacteria are currently a research hotspot for application in the fermented meat industry. Its growth metabolism and derivative metabolites formed during the fermentation of meat products regulate protein degradation and oxidation to a certain extent and improve product quality. Therefore, this paper mainly reviews the changes occurring in proteins in fermented meat products and their effects on the quality of the products. Referring to studies on the effects of lactic acid bacteria on protein degradation and oxidation from all over the world, this review aims to provide a relevant reference for improving the quality of fermented meat products.

## 1. Introduction

Fermented meat products refer to meat products with a long shelf life, special flavor, texture, and color formed through fermentation of raw meat with the natural or artificial addition of microorganisms under specific temperature and humidity conditions [1]. Due to the differences in customs, raw materials, recipes, and manufacturing processes in different countries and regions, there are diverse products in the fermented meat products market [2], such as sausages that originate in the Mediterranean region and have spread to other countries such as Germany, the United States, and Hungary, which have created products with different characteristics due to the differences in each region’s own environment and processing methods [2,3]; the dry-cured hams of Jinhua and Xuanwei in China, Iberia and Serrano in Spain, Parma and San Daniele in Italy, and Bayonne in France all have strong regional characteristics [1,3]; in addition, there are also regional specialties such as salami, cured meat and sour fish in China, Thailand and India [1,4,5].

Protein, a molecularly large and structurally complex compound composed of amino acids, is a crucial dry matter component of meat. During fermentation, maturation, and storage of meat products, moderate protein degradation can effectively improve the flavor and nutritional value of fermented meat, but excessive protein oxidation can adversely affect meat quality such as texture, color, and flavor. In addition, oxidative induction of proteins may affect their digestive utilization, reduce the content of essential amino acids, reduce the nutritional value of meat products, and even cause product spoilage and deterioration, thereby severely affecting the safety of the product for consumption [6,7]. Therefore, we need to determine the optimal degree of protein changes to prevent excessive protein oxidation, thus maximizing the quality and nutritional value of fermented meat products through protein changes.

Lactic acid bacteria (LAB) are dominant bacteria in meat product fermentation. During fermentation, dynamic changes in the flora and the metabolites produced affect the flora structure in the product, act on each meat component, and have a certain impact on the flavor and other product indicators [8,9]. The effect of LAB on the quality of meat products has attracted considerable research attention. Although the research on lactic acid bacteria has developed in recent years, there is still a lack of review on how lactic acid bacteria affect the changes occurring in proteins of fermented meat and further improve the quality of products [10,11,12,13,14]. Therefore, this paper reviews the changes and influence on protein in fermented meat products, the regulation of protein degradation and oxidation by LAB, and its effect on the quality of fermented meat products, aiming to provide a relevant reference for the use of LAB for regulating the moderate change of protein in fermented meat.

## 2. Protein Changes in Fermented Meat Products

### 2.1. Protein Degradation

Protein degradation is among the most crucial biochemical phenomena that occurs during fermentation and maturation of meat products. Degradation of proteins in these fermented meat products is mainly influenced by endogenous enzymes in muscle tissue and proteases produced by microorganisms involved in fermentation. The endogenous proteases can be classified as endopeptidases and exopeptidases. Endopeptidases mainly include cathepsin B, cathepsin L, cathepsin D, and calpain, and exopeptidases mainly include aminopeptidases and carboxypeptidases [15]. Sarcoplasmic protein and myofibrillar protein (MP) were mainly decomposed into polypeptides by endogenous proteases such as cathepsin B and cathepsin L at the early fermentation stage. The polypeptides were further decomposed into small peptides, free amino acids, aldehydes, organic acids, and amines by endogenous aminopeptidases, microbial proteases, and microbial metabolites, which lead to changes in physical and chemical criterion [16]. This is reflected in both fermented sausages and hams; in the early stages of sausage fermentation, although lactic acid produced by microbial metabolism such as LAB decreases the pH of the product, the alkaline metabolites accumulated by protein degradation in the later stages may act as a buffer against organic acids and cause a rebound in the pH of the product [17,18]; in the production of hams, the pH of dry-cured ham was found higher after ageing due to massive release of low-weight nitrogen molecules and ammonia formation ascribed to endogenous and exogenous proteolytic, deaminase and deamidase activities [19].

The protein hydrolysis index, total volatile base nitrogen, and free amino acid content of meat significantly increased during fermentation [20]. The trial of Yu et al. [21] verified that degradation of large molecules of myogenic fibronectin and myosin in fermented sausage results in an increase in the concentration of small molecule peptides. As shown in Table 1, antioxidant peptides in sausage and ham are mainly derived from myosin and actin hydrolysis [22]. Myosin, actin, myoglobin, troponin, and pyruvate kinase are generally the main proteins affecting differential peptides [23]. Using dry cured ham, Li et al. [24] identified 66 peptides from myosin, β-enolase, and actin with antioxidant, angiotensin-converting enzyme, and platelet-activating factor-acetylhydrolase inhibitory activities and found that peptides of <3 kDa were most effective. By measuring dry cured ham at different stages of storage, Virgili et al. [19] verified that free amino acids are released from the ham, biogenic amines are produced as storage time increases and protein degradation continues, and tyramine and putrescine formed due to the conversion of arginine to ornithine were the most abundant at 23 months of storage.

Thus, during the maturation of fermented meat products, large molecule proteins tend to be degraded into small molecule substances; however, excessive degradation can lead to the production of non-healthy factors such as biogenic amines.

### 2.2. Protein Oxidation

Although protein oxidation is similar to lipid oxidation because it is also a chain reaction consisting of initiation, transmission, and termination caused by free radicals [25], the targets and pathways of protein oxidation are more complex and are closely related to the types and properties of oxidation products [26]. Protein oxidation refers to covalent modification of proteins directly induced by active substances such as reactive oxygen species (ROS) and reactive nitrogen species (RNS) or indirectly by secondary products of oxidative stress such as lipids and carbohydrate oxides. The ROS can result in the oxidation of amino acid side chains and protein skeleton, leading to protein fragmentation or protein–protein crosslinking [27]. Among them, the ROS generation system is mainly affected by the radiation, irradiation, and lipid oxidation system; myoglobin-mediated oxidation system; and metal catalytic oxidation system [28].

The possible mechanisms of protein oxidation in fermented meat products are shown in Figure 1. The blue color represents the protein oxidation mechanism catalyzed by metal ions: in stage 1, the hydrogen atom in protein (PH) is deprived of ROS such as a hydroxyl radical (OH**^·^**), thereby generating a carbon-centered protein free radical (P**^·^**); in stage 2, the protein free radical (P) is oxidized to alkyl free radical (POO**^·^**) under the condition of O_2_ [29]; a cross-linked derivative (P-P) was generated by carbon-carbon bonding (C-C) with another protein free radical (P**^·^**) without O_2_; in stage 3, an alkyl radical (POO**^·^**) generates alkyl peroxide (POOH) by seizing the hydrogen atom in another protein, participating in the redox reaction of transition metal ions (M^n+^ such as Fe^2+^ or Cu^+^), or reacting with a protonated superoxide radical; and then, in stages 4 and 5, alkyl peroxides (POOH) react with ROS such as HO_2_**^·^** radical or with reduced forms of transition metal ions (M^n+^) to form alkane oxidation radicals (PO**^·^**), respectively. Further, hydroxyl derivatives (POH) are generated in stages 6 and 7. In addition, alkyl peroxide (POOH) and hydroxyl derivative (POH) can break the peptide chain through α-amidation or the diamide pathway to produce related derivatives [30]. The green color is the speculated mechanism of lipid peroxidation-induced protein oxidation. Similar to protein oxidation catalyzed by metal ions, lipids generate free radicals through the initiation and transmission of oxidation to seize hydrogen atoms in proteins, and then form polymers or protein–lipid complexes through several reaction stages of hydrogen extraction, extension, recombination, and polymerization.

Park et al. [31] compared the sensitivity of common amino acids in meat to the oxidation reactions of different catalytic systems, and confirmed that the amount of essential amino acids such as cysteine, methionine, and tyrosine significantly reduced in metal ion-catalyzed oxidation. This is mainly due to the generation of carbonyl derivatives of these amino acid side chains and the loss of sulfhydryl groups; therefore, experimental studies often use the carbonyl and sulfhydryl content as a measure of the degree of protein oxidation.

### 2.3. Effects of Protein Changes

#### 2.3.1. Effects on Protein Structure

Myofibrillar protein is the main protein in meat, accounting for 55–60% of the total and is mainly composed of myosin, actin, and other regulatory proteins. Many myofibrils form muscle fibers and further connect through the connective tissue to form muscles [32]. Both protein degradation and oxidation during fermentation and maturation change the natural structure of proteins and consequently their properties.

Changes in the protein secondary structure are observed using Fourier transform infrared spectroscopy, circular dichroism spectroscopy, and Raman spectroscopy; UV absorption spectroscopy or endogenous fluorescence spectroscopy are used to detect the tertiary structure of proteins. The relaxation time and proportion of different water are observed through NMR spectroscopy. Thus, we can speculate that changes occur in protein during the production of fermented meat products.

Through Raman spectroscopy, some scholars analyzed that oxidation expands the protein structure in meat and transforms α-helix into β-sheet, β-turn, and random coil [33]. The increase in the hydrophobicity of a protein surface indicates that oxidation exposes hydrophobic amino acids. The change in the spectral wavelength can be used to modify, expose, or cover the oxidation of aromatic amino acid residues, and finally reveal the changes in secondary and tertiary structures of proteins [34]. Lipid oxidation-induced protein oxidation affects the structure and functional properties of protein gels through the expansion and crosslinking of MP. Moderate MP oxidation leads to protein crosslinking through disulfide bonds to maintain a good water-holding capacity of gels. However, excessive exposure of hydrophobic residues and formation of protein polymers during excessive MP oxidation hinder the water-holding capacity of gels [35].

Thus, we can consider that a protein’s structure is directly related to the texture of fermented meat products, and ultimately reflects the hardness, chewing type, elasticity, and tenderness of the products. In addition, the protein structure determines the degree of binding to flavor compounds, especially the number of hydrophobic binding sites on the surface of the protein. Thus, structural changes caused by protein degradation and oxidation affect product quality to some extent.

#### 2.3.2. Impact on Product Quality

The effects of protein changes on fermented meat products are often directly reflected through the color, tenderness, flavor, and edible quality of products. As shown in Figure 2, different levels of the changes occurring in proteins have positive or negative effects on quality.

(1) Color gives the first impression about a product to consumers. According to previous studies, 80–90% of the stability of meat color is determined by the content and chemical state of myoglobin [36]. As shown in Figure 2, myoglobin, which gives the meat its purplish red color, can be oxygenated to form bright red oxygenated myoglobin, while these two myoglobins are oxidized to form brown high-iron myoglobin under an anaerobic condition. Excessive accumulation of high-iron myoglobin in fermented meat products during maturation deteriorates meat color. However, high-iron myoglobin can be reduced with reductase to maintain the stability of meat color [37]. Thus, enzymes related to redox regulation of sarcoplasmic proteins are also closely related to meat color [38].

Moreover, light scattering has a certain influence on meat color, especially meat brightness [38]. By studying dry cured ham, some scholars have speculated that protein degradation and oxidation affect the muscle tissue microstructure and change the distance between muscle filaments, thereby affecting the absorption, transmission, and scattering of light by the muscle and thus changing the color of meat [39,40]. In summary, changes in myoglobin content and state, reductase activity, and muscle structure are critical factors affecting the color of the meat product.

(2) Tenderness, as a crucial index for evaluating meat grade, is determined by many factors. In general, the type of muscle, integrity of muscle fiber, and activity of endogenous enzymes in muscle are all related to muscle tenderness [41]. As shown in Figure 2, protein degradation affects meat tenderness by degrading numerous MPs in muscle tissue [42]. In addition, ROS-induced protein oxidation may damage the MPs and activate the proteasome, thereby promoting the degradation of structural proteins in muscle and improving meat tenderness [43].

However, some researchers have pointed out that over-oxidation of proteins also adversely affects tenderness. Strong oxidation often leads to the formation of cross-linked MPs, loss of thiol groups, and formation of Schiff bases and disulfide bonds, and thus, the activity of calpain is reduced [44]. These cross-linking compounds also increase the distance between muscle fibers, and the non-flowable water in myofibrils is transformed into free water that can be easily lost, thereby reducing the water-holding capacity of meat [45], and resulting in increased hardness of meat and decreased tenderness [7,46].

Based on proteomics, Zhu et al. [47] found that tenderness is closely related to muscle structure, energy metabolism, heat shock protein, oxidative stress response, and apoptosis-related pathways. Thus, moderate protein degradation can improve meat tenderness, whereas excessive oxidation can reduce it.

(3) Lipid hydrolysis and oxidation are generally believed to be the main methods of producing volatile flavor compounds. However, as shown in Figure 3, protein degradation also has a major impact on flavor changes.

Protein degradation produces free amino acids as a taste substance. In ham, taste amino acids such as glutamic acid and aspartic acid are directly positively correlated with the product’s taste. Alanine is related to sweetness, lysine is related to a mature taste, and tryptophan is related to a salty taste. Phenylalanine and isoleucine contribute to an acidic taste. Tyrosine and histidine have very low taste thresholds, which may also affect the flavor of ham [39,50]. Microbial transaminases can convert free amino acids into α-keto acids, which can be directly or indirectly metabolized into corresponding aldehydes as intermediates. Aldehydes are converted to corresponding alcohols and acids by alcohol dehydrogenase and aldehyde dehydrogenase, respectively. Microorganisms are also an indispensable part of flavor substance production by amino acids. In the presence of bacteria such as LAB and Staphylococcus, branched-chain amino acids are converted into corresponding methyl branched-chain compounds such as 2-methylbutyraldehyde, 2-methylpropanal, 3-methylbutyraldehyde, and benzaldehyde, contributing to the malt, fruit, and sweet tastes of the product [14,51]. Estévez et al. [52] reported the formation of Strecker aldehyde from leucine and isoleucine through α-amino aliphatic semialdehyde and γ-glutamic acid semialdehyde. Thus, carbonyl derivatives were believed to be involved as precursors in the degradation of Strecker aldehyde to produce volatile flavor compounds.

On the other hand, although protein itself is tasteless, some researchers believe that it can contribute to product flavor by binding with flavor compounds [53]. Studies have shown that the interaction between proteins and flavor substances depends on physical binding, chemical interaction, and mass transfer effects. For example, hydrogen bonds and van der Waals forces are the key driving forces for the interaction between myosin and ethyl octanoate [54], and hydrophobic forces are also crucial for maintaining the interaction between protein and flavor substances [55]. Notably, ketone and aldehyde are more difficult to bind to the MP gel with an increase in the carbon chain length. By contrast, in the protein solution system, the ability of flavor compounds to bind to proteins increases with an increase in the carbon chain length, which may be because the gel structure hinders the mass transfer of flavor compounds and affects their release [56].

However, excessive protein degradation leads to excessive accumulation of bitter amino acids and peptides such as hypoxanthine, conferring bitterness and rancidity to the products [48]. Excessive oxidation of proteins leads to changes in protein conformation, reduces the hydrophobicity of the protein surface, and reduces the binding sites between proteins and flavor compounds, thereby affecting the ability of proteins to bind flavor compounds [55]. Moderate protein degradation and oxidation enrich the products’ flavor through the Strecker degradation reaction, but excessive protein changes can lead to flavor deterioration.

(4) The edible quality of products, especially safety, is a major concern for consumers. As shown in Figure 2, moderate protein degradation can improve the edible quality of products, but excessive protein oxidation poses a threat to the quality, digestibility, and even, safety of products. Oxidation of meat protein leads to the oxidative modification of amino acid side chain groups, reduction of essential amino acids, and generation of irreversible carbonyl compounds, which thereby reduce the edible value of meat [57]. Ma et al. [58] proposed that during the processing of air-dried yak meat, the digestibility of MPs decreases with an increase in protein oxidation, which may be because the over-oxidation modification of proteins hinders protein hydrolysis, weakens the binding of proteolytic enzymes to proteins, affects the sensitivity of proteins to pepsin, and reduces protein digestibility [59]. Moreover, excessive protein oxidation can directly lead to meat corruption and the formation of derivatives (such as furans, biogenic amines, acrylamide, and nitrosamines), thus causing food safety problems [60].

Therefore, to ensure the quality of fermented meat products and to improve this quality, the degree of protein changes must be adequately controlled. At present, the degree of protein degradation or oxidation is regulated by adding NaCl, auxiliary additives, and fermenting agents and by controlling the production temperature.

## 3. Regulatory Role of LAB

As organisms generally recognized as safe, LAB have become the most widely used probiotics in food production. Currently, *Lactobacillus lactis*, *Lactobacillus plantarum*, *Pediococcus lactis*, and *Pediococcus pentosaceus* and so on are widely used in fermented meat products [61,62]. They regulate the degradation or oxidation of protein mainly through acid and protease production, and their antioxidant capacity.

### 3.1. Acid Production

Endogenous enzymes play a major role in protein degradation during meat product fermentation; however, LAB contribute to the degradation of muscle protein and the release of small peptides and amino acids through metabolic acid production. Studies have shown that some endopeptidases, such as cathepsin, exhibit higher activity at low pH, and increasing pH inhibits cathepsin B activity to a certain extent [63,64]. By comparing fermented sausages under the two acidification curves, Berard et al. found that the number of actin-derived peptides identified at low pH increased from 42 to 144, which was 88 more than that identified at high pH. Most of these derivatives were released at the cleavage sites of cathepsin B and cathepsin D. This shows that cathepsin can degrade proteins to produce more peptides under acidic conditions [65].

The LAB can produce organic acids such as lactic acid through carbohydrate metabolism in fermented meat, resulting in a decrease in environmental pH [66]. Therefore, the acid production capacity is often used as one of the criteria for measuring the suitability of a LAB starter culture. Fermented sausages experience a decline in the pH value at the beginning of fermentation. The pH decline added with a LAB starter is more obvious, which thus provides a better environment for protease activity. At the late stage of fermentation, the pH of sausages show a certain rebound, which is mainly due to protein degradation by protease and the production of peptides, amino acids, amines, and organic acids [62]. By analyzing a protein degradation model at pH 4.0, Fadda et al. [67] found that the degradation under acidic conditions was affected by the synergistic effect of acid-induced changes by acidic endogenous protease, protease hydrolysis activity of *L. plantarum* CRL 681, and fermentation metabolism. Therefore, LAB can be speculated to assist in the initial degradation of proteins by producing acid and reducing pH to improve cathepsin activity [68]. In addition, the metabolism and acid production of LAB will affect the structure of other microorganisms, thus inhibiting the growth of spoilage bacteria, and reducing the impact of spoilage bacteria metabolism [69].

### 3.2. Protease Production

Protein degradation is mainly affected by endogenous protease, but LAB-secreted protease during fermentation can also increase the concentration of free amino acids and short peptides by affecting the degradation of sarcoplasmic proteins [70].

Compared with dairy products, the research and application of lactobacillus protease in fermented meat are less. On screening LAB isolated from fermented sausages, Cao et al. [10] obtained seven strains with good fermentation performance and high protease activity, and identified one strain as *L. pentosus*, one strain as *L. fermentum*, and the others as *L. plantarum*. Through SDS-PAGE and molecular docking, Sun et al. verified that proteases purified from *P. pentosaceus* R1 [71], *L. brevis* R4 [72], *L. curvatus* R5 [73], and *L. fermentum* R6 [74,75] isolated from Harbin dry sausage can catabolize sarcoplasmic proteins and myogenic fibrous proteins, and interact with the myosin light chain, myosin heavy chain, and actin and myoglobin in meat. They also predicted that protease consists of a β-strand and an α-helix, the properties of protease are altered by changes in a tandem conserved structural domain consisting of a protease structural domain and a PDZ structural domain, and the active site of protease is related to protease properties and substrate specificity [71]. This demonstrates the potential of protease-producing LAB to influence proteolysis as a meat fermenter.

In addition, Cao et al. [10] inoculated LAB with high protease activity into fermented sausage and identified the metabolites of sausage protein through LC-MS/MS. *Lactobacillus plantarum* CD101 could help to produce more peptides and small peptides, and the content of antioxidant peptides also significantly increased, indicating that LAB has a certain application prospect as a meat fermentation starter.

### 3.3. Antioxidant Properties

The LAB are considered a natural antioxidant, as shown in Figure 4. At present, the antioxidant properties of LAB are believed to be jointly regulated through multiple mechanisms such as regulation of the redox system, production of antioxidant metabolites, scavenging of free radicals, and chelation of metal ions [12].

First, transition metal ions inhibit the enzyme-catalyzed phosphate substitution reaction and produce related ROS by decomposing hydrogen peroxide. The cell-free extract of LAB cells exhibited a good chelating ability of Fe^2+^ and Cu^+^. Therefore, LAB cells are speculated to contain related metal ion-chelating agents and their high antioxidant activity is positively correlated [76,77].

Second, several studies have shown that the intracellular antioxidant activity of LAB is based on the activities of superoxide dismutase (SOD), catalase (CAT), glutathione peroxidase (GPx), nicotinamide adenine dinucleotide (NADH) oxidase, NADH-peroxidase, and glutathione (GSH) [78,79,80]. Thus, antioxidant enzymes play an indispensable role in the high antioxidant properties of LAB. The SOD regulates the ROS level by catalyzing superoxide decomposition [81], while CAT and GPx decompose H_2_O_2_ formed by SOD-mediated superoxide anion radical scavenging into water and oxygen, thereby preventing the formation of the hydroxyl radical (·OH). Regulation of some ROS-producing enzymes can also play an antioxidant role [82]. The NADH oxidase produces H_2_O_2_ that causes NADH peroxidase to remove H_2_O_2_ [80]. The thioredoxin–thioredoxin reductase system (Trxs) and glutathione–glutathioredoxin system (GSHs) in LAB cells can regulate intracellular disulfide/disulfide, thus helping to maintain redox homeostasis [83]. Under the premise of hydrogen supply by NADPH oxidase, TrxR converts oxidized Trx into reduced Trx, GR converts oxidized GSH into reduced GSH, and GSH activates GPx activity to remove lipid peroxides and H_2_O_2_ [84].

Furthermore, LAB can produce various metabolites with antioxidant activities, such as cell wall polysaccharides, peptidases, lipid acids, and protein derivatives [85,86]. Exopolysaccharide is a type of polysaccharide produced by LAB during metabolism that adheres to the surface of bacteria and binds to cells [87]. *Lactobacillus plantarum* AAS3 screened from fermented dried fish can scavenge DPPH free radicals and hydroxyl free radicals through metabolism to produce extracellular polysaccharide, thus reflecting a good in vitro antioxidant capacity [88]. Fatty acids such as palmitic acid and stearic acid are considered indirect antioxidants. In saturated fatty acids, the antioxidant activity increases with an increase in chain length, while unsaturated fatty acids are mainly oxidized in an unsaturated-dependent manner [85]. Some amino acid derivatives such as N-(5-amino-2-hydroxy-1-oxopentyl)-tyrosine, (S)-8-hydroxy-4-hydroxy-phenylpropionic acid, (S)-8-hydroxy-phenylpropionic acid, and indole-3-(S)-3-lactic acid can scavenge DPPH free radicals, which proves the antioxidant activity of metabolites of LAB [89].

In addition, in the face of oxidative stress, through proteomics, some researchers analyzed that LAB can inhibit biochemical processes related to the ribosome structure, transcription, translation, and protein synthesis through the oxidation defense system, oxidation protein repair system, ABC transporter system, and the expression of proteins involved in carbohydrate energy metabolism, and carry out antioxidant stress reactions [90].

Mei et al. [11] added *L. plantarum* P3 with antioxidant properties to fermented sausage, and detected the thiobarbituric acid value, carbonyl content, thiol content and other indicators. The P3 could inhibit the activity of lipoxygenase and prevent adverse consequences caused by lipid peroxidation by reducing lipid oxidation through its antioxidant properties and chelating with ferrous ions, which verified the slowing effect of P3 on protein oxidation of sausage. Scavenging free radicals, chelating metal ions, and preventing lipid peroxidation are the ways through which LAB with antioxidant function reduce protein oxidation and improve the quality of fermented meat.

### 3.4. Effect on Quality

People currently want to develop more green, safe, healthy, and delicious functional foods. The safety of fermented meat products is the focus of consumers. As a key factor leading to spoilage and deterioration of products, LAB can compete with other microorganisms for nutritional substrates in fermented meat products and metabolize them to produce lactic acid, acetic acid, hydrogen peroxide, antimicrobial peptides, and other compounds. These compounds can destroy the stability of biofilms of spoilage bacteria through coordination, interfere with the proton gradient of spoilage bacteria, passivate the enzyme activity of spoilage bacteria and generate ROS after entering cells, inhibit the growth and metabolism of spoilage bacteria, and improve product safety [69]. They can also reduce the residual amount of nitrite through acid degradation or enzyme degradation and reduce the potential harm caused by nitrite [91]. At the same time, LAB can improve the color by increasing the brightness value of the product [13]. Some special strains can convert myoglobin and high-iron myoglobin (Met-Mb) to red nitro myoglobin (Mb-NO), with the potential of natural biological colorants [92]. LAB-produced enzymes can promote the decomposition of protein lipids, which is closely related to the production of esters, aldehydes, and ketones [8,14].

Moreover, LAB can maintain intestinal stability, coordinate digestion, regulate cholesterol content [93], and regulate Nrf2-Keap1-ARE [94], NFκB [95], and MAPK [96] signaling pathways through antioxidant characteristics to alleviate oxidative stress, improve inflammation, reduce toxins, and alleviate injury.

Therefore, based on the functional characteristics of metabolic acid production and protease production, and antioxidant properties, LAB can be used as a starter culture for regulating the degree of protein changes in fermented meat products, ensuring the quality of the products, and further assisting in the development of green and healthy products.

## 4. Summary

Protein changes occur through the whole process of fermentation, ripening, and storage of fermented meat products. In this paper, the mechanism of protein degradation and oxidation, its effect on the quality of fermented meat products, and the regulation of protein degradation and oxidation by LAB are described in detail. The use of LAB as a starter culture for meat fermentation can not only improve sensory qualities such as color, tenderness, and flavor of fermented meat products but also affect the degree of protein degradation and oxidation in the products through acid production and protease production, and antioxidant properties. However, few studies have currently investigated whether the addition of LAB can affect the protein structure and improve the binding ability of proteins to various flavor substances. Therefore, studies on LAB starters are warranted.

With the advancement of omics technology and computer simulation software, in the future, we can use multi-omics technology combined with molecular docking and other models to clarify the dynamic process of the changes occurring in proteins of fermented meat, determine the key differential metabolites of proteins that affect product quality, and identify how to maintain the best quality of products. The effects of LAB or their interactions with other influencing factors on each component in fermented meat were clarified to optimize the LAB starter, explore the improvement mechanism of strain compounding on fermented meat products, and develop more high-quality fermented meat products.

## Figures and Tables

**Figure 1 foods-11-02427-f001:**
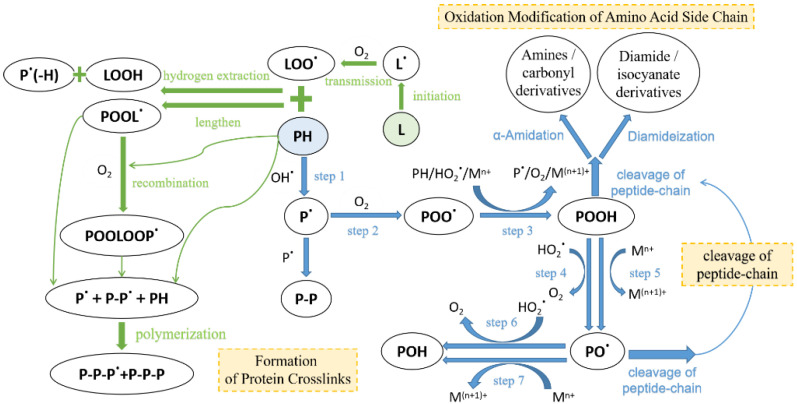
Mechanism of Protein Oxidation in Fermented Meats [7,25]. Notes: P = protein; L = lipid. The blue color represents the metal ion-catalyzed protein oxidation mechanism, and the green color represents the lipid peroxidation-induced protein oxidation.

**Figure 2 foods-11-02427-f002:**
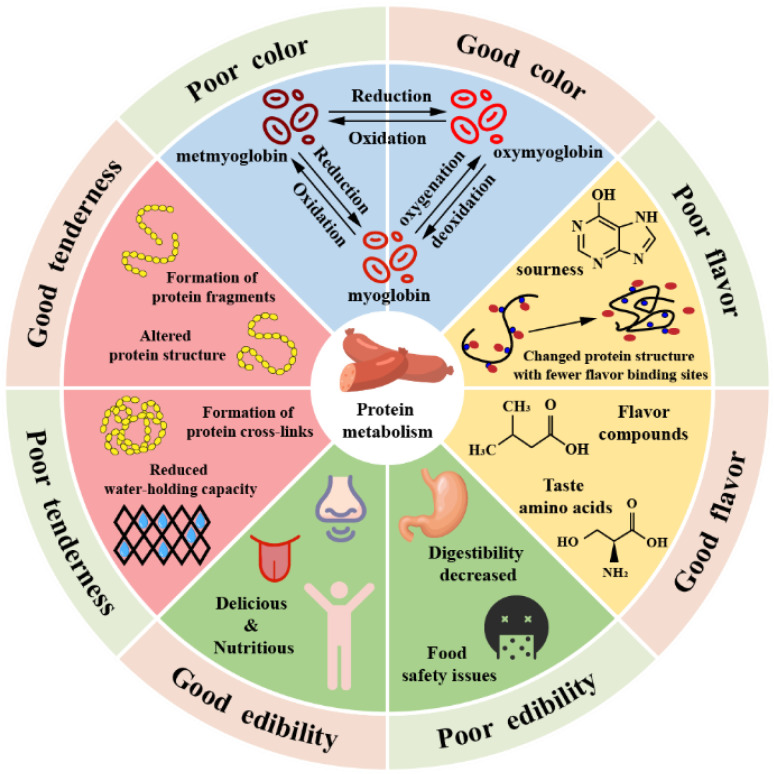
Effects of Protein Changes on the Quality of Fermented Meat Products.

**Figure 3 foods-11-02427-f003:**
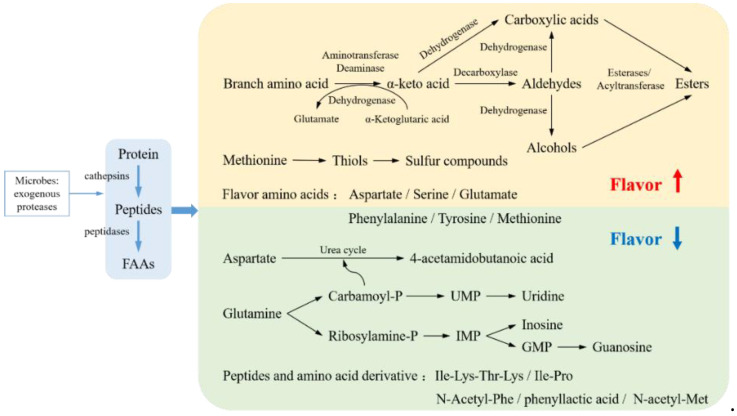
Effects of Protein Degradation on the Flavor of Fermented Meats [48,49].

**Figure 4 foods-11-02427-f004:**
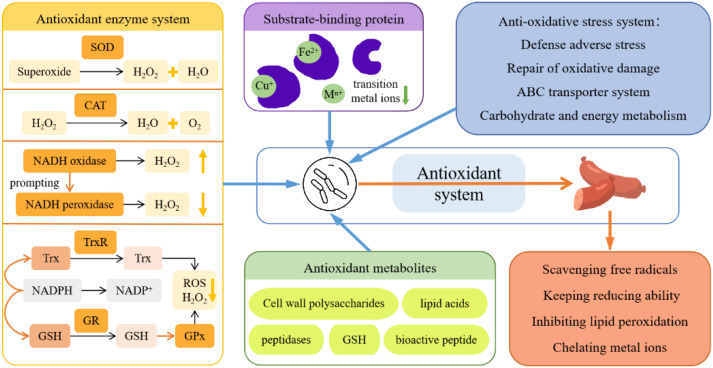
Antioxidant regulation mechanism of LAB. Note: SOD: superoxide dismutase; CAT: catalase; NADH: nicotinamide adenine dinucleotide; Trx: thioredoxin; TrxR: thioredoxin reductase; NADPH: nicotinamide adenine dinucleotide phosphate; GR: glutathione reductase; GSH: glutathione; GPx: glutathione peroxidase.

**Table 1 foods-11-02427-t001:** Composition of Crude Peptide in Fermented Meat.

Sample	Peptide Sequence	Length	Mass (Da)	Protein of Origin	References
Fermented sausage	LPVKY	5	618.3741	Phosphatidylinositol 3-kinase catalytic subunit type 3	[22]
	PFGDTH	6	672.2867	CDP-6-deoxy-delta-3,4-glucoseen reductase	
	QPSLVH	6	679.3653	Aminoglycoside transferase	
	LPVTVR	6	683.433	Rho guanine nucleotide exchange factor 2	
	LPSLKF	6	703.4268	Phosphoglycerate kinase	
	HKLPVK	6	720.4646	Myoglobin	
	YGLDEK	6	723.3439	Serine/threonine-protein kinase A	
	LDRKDL	6	758.4286	Lipid A biosynthesis lauroyltransferase	
	LDLRDK	6	758.4286	Tetratricopeptide repeat protein 27	
	EGGLSKLTLDKLDVKGK	17	1800.036	Phosphoglycerate kinas	
	EARSKLTLDKLDVKGK	16	1800.047	Phosphoglycerate kinase	
	ESDLLAAFR	9	1020.524	Adenosine kinase	
	EVSHGSDQVKAHGQK	15	1605.786	Hemoglobin subunit alpha	
	ELSNKLTLDK	10	1159.645	Phosphoglycerate kinase	
	VDLNGGSHAGNK	12	1167.563	Beta-enolase isoform X1	
	EGADSEMALFGEAAPYLRKSEKEVGKN	27	2925.418	Myosin-1 isoform X1	
	NPPKF	5	601.3224	Myosin	
	KGGSLELTLDKLDVKGK	17	1800.036	Phosphoglycerate kinase	
Mutton Ham	MWTD	4	551.61	Uncharacterized protein	[23]
	SAGNPN	6	558.20	Uncharacterized protein	
	APYMM	5	611.76	Uncharacterized protein	
	VFDPEG	6	662.69	Myosin-2	
	FWIIE	5	706.84	Spectrin alpha chain-like protein	
	DKEEFV	6	744.81	Myosin-7	
	GVDNPGHP	8	791.81	Creatine Kinase M-type	
	MVHMASK	7	803.00	Glyceraldehyde-3-phosphate dehydrogenase	
	PAPAPPKE	8	805.91	Titin	
	FGNTHNK	7	816.87	Creatine Kinase M-type	
	MDAIKKK	7	833.05	Tropomyosin	
	IEEALGDK	8	873.94	Beta-enolase	
	IPPKIPEGE	9	887.01	Troponin T	
	GLRKHER	7	895.02	Myosin-4	
	ERFSKDE	7	909.93	Uncharacterized protein	
	ERSFKDE	7	909.93	Troponin T	
	PFGNTHNK	8	913.99	Creatine Kinase M-type	
	NVINGGSHAG	10	924.97	Glyceraldehyde-3-phosphate dehydrogenase	
	HIITHGEE	8	935.02	Myosin regulatory light chain 2	
	GRKFRNPK	8	1002.18	Beta-enolase	
	DVAGHGQEVL	10	1024.10	Myosin regulatory light chain 2	
	IDDMIPAQK	9	1030.20	Creatine Kinase M-type	
Jinhua Ham	GKKFNV	6	565.28	Transcription activator BRGl	[23]
	LVVDGVK	7	728.88	Uncharacterized protein	
	DKEEFV	6	744.81	Myosin-7	
	VDIINAK	7	772.45	Uncharacterized protein	
	VHMASKE	7	801.39	Glyceraldehyde-3-phosphate dehydrogenase	
	MDAIKKK	7	833.05	Tropomyosin	
	LVVDGVKL	8	841.53	Creatine Kinase M-type	
	IPPKIPEGE	9	887.01	Troponin T	
	KAGTTPKGK	9	887.04	Maturase K	
	YGEKLKR	7	893.04	Binding protein	
	ALPHAIMR	8	908.11	Actin	
	PFGNTHNK	8	913.99	Creatine Kinase M-type	
	VKQKGPDF	8	918.50	Pyruvate kinase	
	AGQAFRKF	8	924.05	Uncharacterized protein	
	MVHMASKE	8	932.43	Glyceraldehyde-3-phosphate dehydrogenase	
	IEEALGDKA	9	945.02	Beta-enolase	
	VITHGDAKD	9	955.48	Myosin regulatory light chain 2	
	LQNHPEHS	8	960.99	Glyceraldehyde-3-phosphate dehydrogenase	
	EAGPSIVHR	9	965.07	Actin	
	LRDKAKEL	8	972.12	Binding protein	
	AGFAGDDAPR	10	975.99	Actin	
	FPMNPPKF	8	977.49	Myosin	
	DVGDWRKN	8	989.05	Troponin T	
	VAPEEHPTL	9	992.08	Actin	
	KKAGTTPKGK	10	1015.21	Troponin T	

## Data Availability

Not applicable.

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
