# Peer review of "The Changes Occurring in Proteins during Processing and Storage of Fermented Meat Products and Their Regulation by Lactic Acid Bacteria"

_foods, 2022, doi:10.3390/foods11162427_

Round 1

Reviewer 1 Report

Raw sausages are a group of meat preparations characterised by a specific taste and odour profile, a red cross-sectional colour and a relatively high shelf life and nutritional value. They are made from raw meat, mainly pigmeat and beef, and pig fat (backfat). In order to achieve the typical product and organoleptic characteristics desired for the respective assortment, they undergo maturation, i.e. they undergo fermentation and drying under different conditions. These include time, temperature, relative humidity, speed of air exchange (movement) and smoking with cold smoke. Changes in the filling components (physical, chemical, biological) shape the sensory properties of the final product and its durability. Raw sausages were already produced in ancient times, when salting, fermenting, drying or drying the sausages were the only methods of preservation. Fermentation took place spontaneously using the natural microflora of the meat, the spices and the environment. The special taste and smell profile of raw sausages is the result of the transformation of proteins, fats and carbohydrates from the filling. The dark red colour on the cross-section of the product is formed by the transformation of the dyes and their concentration during the maturation/drying of the product. The own carbohydrates of the meat (glycogen) and the raw sausages added to the filling of raw sausages are homo-fermented or hetero-fermented both during production and storage. This transformation can take place under – anaerobic conditions (e. g. milk or alcohol fermentation), – aerobic conditions (e. g. oxidation in the cancer cycle or in the pentosan cycle, oxidative decarboxylation of pyruvinic acid).

The review article “Protein Metabolism and its Effects on Fermented Meat Products and its Regulation by Lactic Acid Bacteria” seems to be an interesting piece of literature. According to the reviewer, it provides a factually well-prepared explanation and writing of the mechanisms of oxidation of protein. The interesting and clear graphical representation of the changes taking place should be highlighted. Interesting and at the same time on a high scientific level is the description of the Regulation mechanism of antioxidation of LAB. I very much appreciate the combination of the theoretical considerations presented with the practical aspect of the effects of the phenomena described on product quality, including colour, fragility and taste. The explanations are based on numerous literary entries from the last period. As already mentioned The specific taste and smell profile of raw sausages is the result of the transformation of proteins, fats and carbohydrates of the filling. I hope the authors will prepare an equally interesting publication on the conversion of fats and carbohydrates.

Below are comments that should be referred to and possibly corrected in the text:

• In the text (line 116) appears “reduced transition metal (Mn+) “, which is the transition metal? Or just iron? Or others (e. g. Cu or Ni)? And if someone else, how did he get there? Was it part of an enzyme?

• Write access (line 116) should be Mn+. Text notation (Mn+) means a positive mangancation – according to the reviewer this is invalid (shown correctly in the diagram)

• According to the reviewer, a transition metal ion should be reduced and not a transition metal (line 116).

• In the English nomenclature, the radical theorem has a point in the upper index like the charge, e. g. P· and not as in the text (line 116).

Reviewer 2 Report

Dear Authors

The proposed review is relevant to modern meat science.  Meat and meat products compared with other food products have a very subtle complex aroma and flavor, which under the influence of technological, biochemical, microbiological and other factors can change in a desirable or undesirable direction. The article describes in detail the influence of lactobacilli on the physico-chemical properties, functional and sensory characteristics, as well as on the taste and aroma of meat by deep analysis of the processes occurring during protein metabolism. The use of starter cultures in meat production is a modern trend for obtaining new types of meat products with high nutritional value and specific organoleptic indicators. The review is well structured, with separate subsections describing protein degradation and oxidation, role of LAB on different quality properties of fermented meat.

Please correct the title  Its sounds not properly (two times used  the word "its")

Line 52:  need citations when you mention "few studies"

I suggest to add one paragraph on  changes of  pH  during protein degradation

Reviewer 3 Report

The article presents a review regarding the changes occurring in proteins during processing and storage of dry fermented meat products. The article does not deal with protein metabolism, as the authors point out, which means that the title of the article does not correspond to its content.

Moreover, the article was poorly organized. The part of the article that the authors refer to as "4. Discusion" is rather a Summary.

Another serious problem is the wrong way of referring to literature, inconsistent with the requirements of the journal, eg line 83 authors write "R. Virgili [13]" instead of "Virgili et al. [13]".

The article should be checked by a native speaker, because errors that appear in its content often make it difficult to understand the meaning, eg lines 23-24 What means "protein metabolism at home and abroad"?

Lines 33-35 - a reference to literature should be given, the more so as the truth of these sentences raises doubts. The first sentence is not true as different types of fermented meat products, such as fermented neck, are produced in different regions. What did the authors mean in the second sentence "domestic market", Chinese?

Many similar problems appear in the article. All this means that, in my opinion, the quality of the article is not appropriate.

Round 2

Reviewer 3 Report

I appreciate the authors' efforts, although the article has still not been improved in many places. In the first review, I pointed to the problem with the wrong way of referring to literature. I have given an example, although I mentioned that the problem is with the entire manuscript. Unfortunately, the authors did not check and correct the entire article, so similar errors can be found in many places, line 100, 105, 341, 395, etc.

Line 347 - figure 3 or figure 4?
